# Investigation of a Possible Relationship between Anthropogenic and Geogenic Water Contaminants and Birth Defects Occurrence in Rural Nebraska

**Balkissa S. Ouattara [1,\*], Muhammad Zahid [1], Farzana I. Rahman [2,3], Karrie A. Weber [2,3,4,5], Shannon L. Bartelt-Hunt [6] and Eleanor G. Rogan [1]**

[1] Department of Environmental, Agricultural and Occupational Health, College of Public Health, University of Nebraska Medical Center, Omaha, NE 68198-4395, USA; balkissouatt@gmail.com (B.S.O); mzahid@unmc.edu (M.Z.); egrogan@unmc.edu (E.G.R.)

[2] Daugherty Water for Food Institute, University of Nebraska, Lincoln, NE 68588, USA; frahman2@huskers.unl.edu (F.I.R.); kweber@unl.edu (K.A.W.)

[3] Department of Earth and Atmospheric Sciences, University of Nebraska-Lincoln, Lincoln, NE 68588-0340, USA

[4] School of Biological Sciences, University of Nebraska-Lincoln, Lincoln, NE 68588-0118, USA

[5] Child Health Research Institute, Omaha, NE 68198, USA

[6] Department of Civil and Environmental Engineering, College of Engineering, University of Nebraska-Lincoln, Lincoln, NE 68588-0531, USA; sbartelt@unl.edu

\* Correspondence: balkissouatt@gmail.com

**Abstract:** Relatively high concentrations of anthropogenic (atrazine and nitrate) and geogenic (uranium and arsenic) water contaminants have been found in drinking water in rural Nebraska. This research assessed a potential association between birth defects occurrence and the contaminants mentioned above within selected Nebraska watershed boundaries. The prevalence of birth defects and the mean concentrations of the selected water contaminants were calculated. More than 80% of Nebraska watersheds had birth defect prevalences above the national average (5 cases per 100 live births). In the negative binomial regression analysis, a positive association was observed between higher levels of nitrate in drinking water and the prevalence of birth defects. Similarly, compared to watersheds with lower atrazine levels, watersheds with atrazine levels above 0.00 μg/L had a higher prevalence of birth defects. This study suggested that chronic exposure to the selected waterborne contaminants even below the legislated maximum contaminant levels may result in birth defects. It also highlighted the relationship between anthropogenic activities (agriculture practices), water contamination, and adverse health effects on children. An additional cohort study is recommended to support these findings so that regulations can be implemented in the form of continuous monitoring of water in private wells and improvements to agricultural practices.

**Keywords:** birth defect prevalence; atrazine concentration; nitrate concentration; agrichemicals in drinking water; uranium concentration; arsenic concentration

## 1. Introduction

Birth defects, also known as congenital malformations, are functional or structural anomalies that can affect any body part(s). They happen during pregnancy and are detectable in utero, at birth, or later during infancy [1,2]. Birth defects annually affect approximately 3% of all babies (1 in 33 babies) born in the United States (US) [1]. Between 2011 and 2015, about 6% (2 in every 33 births) of all babies born in Nebraska had at least one birth defect [3].

Birth defects account for 20% of all infant deaths and represent the leading cause of infant deaths in the US [4]. Moreover, birth defects are costly. In 2004 there were almost

140,000 hospitalizations for the care of infants with birth defects, which represented 2.6 billion dollars in hospital costs [5].

Causes for birth defects remain poorly understood. A US population-based study found that a cause was established in only 1 out of 5 birth defect cases [6]. Unknown etiologies were the leading cause of birth defects (79.8%), followed by genetic causes and environmental teratogens. Many researchers think that most birth defects occur due to multiple factors, including genetic, behavioral, and environmental components [6]. Some factors are described as being associated with the occurrence of birth defects; they include prenatal smoking and substance use (alcohol, drugs), obesity, uncontrolled diabetes, infections, and advanced maternal age [7,8].

Similarly, exposure to some anthropogenic environmental toxicants has been linked to birth defect occurrences. Among these anthropogenic contaminants are atrazine and nitrate, which are widely used in agriculture for pest control and fertilization, respectively. Maternal atrazine exposure was associated with male genital malformations (hypospadias, cryptorchidism, and small penis), choanal atresia and stenosis in offspring [9], and gastroschisis [10]. Maternal consumption of drinking water bearing relatively high nitrate concentrations was associated with general birth defects [11], and specific defects such as neural tube defects [12,13] and limb deficiencies [14].

The teratogenic potential of nitrate is supported by the formation of *N*-nitrosamines in acidic conditions when nitrite reacts with amines. It is suggested that *N*-nitrosamines can cause malformations of the embryo or fetus [15].

Agricultural activities are extensively practiced in rural Nebraska [16] with the intensive use of agrichemicals [17–19]. These agrichemicals (nitrate and atrazine) can contaminate local waterways and groundwater, both of which serve as drinking water [20]; furthermore, it was elaborated that maternal exposure to nitrate and atrazine were associated with birth defects. In rural Nebraska, drinking water is mainly sourced from groundwater, which is supplied by private wells that are not regulated or monitored for contaminant concentrations [21]. Nonetheless, the maximum contaminant limit (MCL) established by the United States Environmental Protection Agency (EPA) under the Safe Drinking Water Act (SDWA), was set at 10 mg/L for nitrate and 3 μg/L for atrazine [22]. Additionally, agricultural activities can lead to the mobilization of geogenic (naturally occurring) elements from rocks, sediments, and minerals, resulting in the contamination of drinking water sources [23]. For example, elevated concentrations of nitrate, near the MCL, have been correlated with the mobilization of naturally occurring uranium in the High Plains aquifer [24,25]. Moreover, moderate to high arsenic concentrations have been found in Nebraska [26]. Studies have shown that uranium [27–29] and arsenic [12,30–32] are potential risk factors for birth defects.

Thus, the present research was conducted to investigate the relationship between birth defects and atrazine, uranium, arsenic, and nitrate concentrations in water, in order to determine whether the reported excess risk of birth defects in Nebraska compared to the nation is related to the overuse of the agrichemicals mentioned above. The four contaminants were chosen because, as described earlier, water contamination at higher concentrations with nitrate, atrazine, uranium, and arsenic is frequent in Nebraska, and these contaminants are mostly related to the agricultural practices in the state.

The unique approach of this research is to focus the environmental contamination and birth defect prevalence on watershed delineations rather than on traditional geographic census entities such as census tracts, zip codes, and counties. This approach considers the transport pathways of waterborne agrichemicals, which do not follow census boundaries but instead travel to local waterways that flow downstream within specific watersheds after precipitation or irrigation events, and can subsequently leach to groundwater [33].

## 2. Materials and Methods

### 2.1. Case Definition, Study Population and Site, and Data Sources

2.1.1. Study Sites

Nebraska counts 23 natural resource districts (NRD) that are set up along hydrological boundaries, allowing them to undertake natural resource management on a watershed basis [34]. This study was conducted within two NRDs, the Lower Elkhorn NRD and the Upper Big Blue NRD. These two NRDs were selected based on prior research that indicated poor health outcomes for children, which include a high incidence of pediatric cancer [33].

Groundwater samples were collected from domestic (privately owned) wells among the rural population within the Logan, Lower Elkhorn, North Fork Elkhorn, Upper Elkhorn, Upper Big Blue, and West Fork Big Blue watersheds, in coordination with the Natural Resource Districts (NRD) in the Lower Elkhorn and Upper Big Blue located in Eastern Nebraska, and with the consent of study participants (Figure 1). Groundwater within these regions was withdrawn from aquifers that can be directly connected to the watersheds by gaining and losing streams.

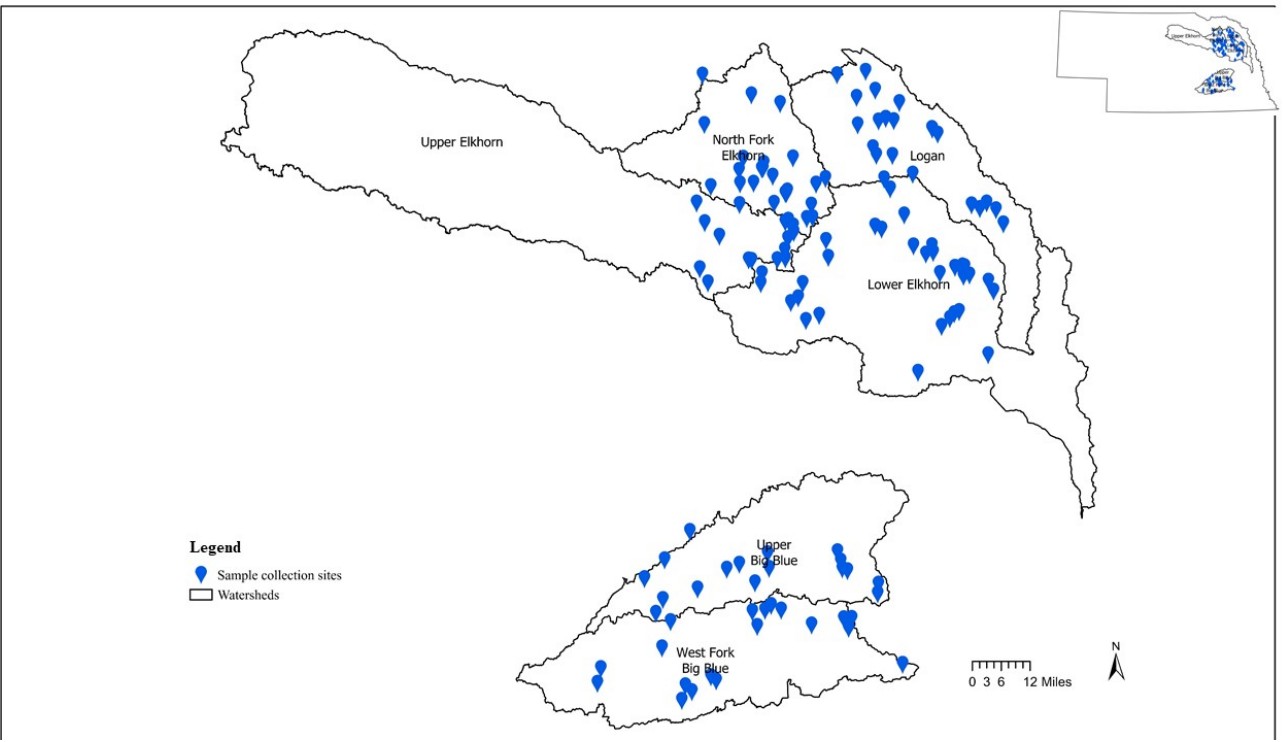

**Figure 1.** Sample collection sites located in Nebraska. Blue points denote sample collection locations.

2.1.2. Birth Defect Data

Secondary birth defect data were requested through submission of data use agreement forms and obtained from the Nebraska Department of Health and Human Services (DHHS). They corresponded to all birth defects recorded in the Nebraska Birth Defect Registry from 1 January 1995 to 31 December 2014. The data from the birth defect registry included the mother's address of residence that was used to geocode geographical coordinates—longitude and latitude—for mapping purposes using the ArcGIS World Geocoding Service [Environmental Systems Research Institute (ESRI) [35]. Information on the mother's smoking and diabetes status was also included in the registry.

Cases were defined from the registry as an occurrence of one or multiple birth defects in a child. Thus, when a child was identified with multiple birth defects, each defect was counted as a separate case. This approach was used to calculate the prevalence of defects

at birth. However, for the geospatial analysis, each infant with a birth defect was represented once on the map.

The at-risk population comprised all live births in Nebraska during the study period—1 January 1995 to 31 December 2014. Thus, corresponding live-birth data were obtained from the Nebraska DHHS. The live birth data set included variables such as the mother's state, city, and zip code of residence. This information was used to obtain the geographical coordinates related to the mother's location for mapping purposes.

### 2.1.3. Water Quality Data

Groundwater samples were collected from domestic wells between June 2021 and February 2022 by the NRDs. Samples were collected for the analysis of anthropogenic and geogenic water quality contaminants atrazine, nitrate, uranium, and arsenic. Prior to sample collection, if possible, the screen was removed, and the water was allowed to run for 2–3 min. Each collected sample was assigned a unique identification number. Samples for atrazine analysis were collected in amber glass bottles without a preservative and immediately stored at 4 °C prior to analysis. Samples for nitrate analysis were immediately collected and preserved with sulfuric acid as described by EPA method 300.0. Samples for uranium and arsenic analysis were collected and preserved with nitric acid as described by EPA method 200.8.

Atrazine was determined using an enzyme immunoassay method [36] following EPA SW-846 method #4670. Atrazine groundwater concentrations were determined relative to a standard curve on a photometer with a method detection limit of 0.04 ppb. Nitrate was determined in accordance with EPA method 300.0 for drinking water on a DIONEX-ICS-3000 ion chromatograph using conductivity detection following elution across a AS9-HC-column. The method detection limit was 0.48 mg/L. Uranium and arsenic were determined in accordance with EPA method 200.8 for the determination of trace elements in drinking water using inductively coupled plasma-mass spectrometry (ICPMS) on a Thermo Dionex IC 5000+ iCAP RQ ICP-MS. Indium was used as an internal standard to validate recovery. The detection limits were 0.036 µg/L for arsenic and 0.030 µg/L for uranium.

### 2.1.4. Geographic Information System (GIS) Data

The Nebraska state boundary and hydrologic unit code (HUC) 8 level watersheds boundary shapefiles were retrieved from the Nebraskamap.gov website [37]. The birth defect and water quality data were geocoded in ArcGIS Pro Ver. 2.4 to include geographic coordinates. The join function in ArcGIS was used to represent both the birth defect data and water quality data at the HUC-8 watershed level. The geographic coordinate system that was used corresponded to the North American Datum (NAD) 1983.

### *2.2. Data Analysis*

### 2.2.1. Birth Defect Prevalence

In the current study the prevalence of birth defects was calculated. In ideal conditions, birth defect occurrence is measured by incidence rates instead of prevalence. Incidence rates quantify the occurrence of new cases in a population during a specific period. The formula to calculate the incidence of birth defects is as follows Equation (1) [38]:

$$\frac{\text{Number of new cases of birth defects in an area and period}}{\text{number of conceptions at risk of developing birth defects in that area and period}} \times \text{multiplier} \qquad (1)$$

However, in this formula, the number of conceptions is not known since the number of spontaneous abortions is unknown. Thus, we are not able to appropriately determine the incidence of birth defects. As a result, most professionals addressing birth defects use the concept of "prevalence" for birth defect occurrence.

Birth defect occurrence is measured using the formula for birth at prevalence [38]. The current study used the following formula:

$$\frac{\text{Number of new cases of birth defects in each Nebraska watershed from 1995-2014}}{\text{the number of live births in Nebraska watershed from 1995-2014}} \times 100$$

Watersheds with a total live births less than 100 infants during the study period (1995–2014) were not included in the analysis. Out of the 72 Nebraska watersheds at the HUC-8 level, birth defect prevalence was calculated for fifty-two (52) watersheds. The cutoff of 100 was based on preliminary data analysis that showed an extremely high prevalence for watersheds with live birth counts between 0 and 100 infants.

2.2.2. Analysis of the Association between Birth Defects and Water Quality Data

All analyses were conducted in SPSS (Statistical Package for the Social Sciences) version 28.0 [39]. The outcome variable was the count of infants with at least one birth defect per watershed, with the watershed total live births during the study period as the scale weight variable. The main independent variables were composed of the mean nitrate and atrazine concentrations per selected watershed. Mean arsenic and uranium concentrations as well as diabetes count per selected watershed were used as covariates.

With an outcome variable that is a count variable, the negative binomial regression analysis was appropriately used, and the full model was set as follows Equation (2):

$$\ln(\text{birth defect counts}) = b0 + b1 \times \text{mean nitrate} + b2 \times \text{mean atrazine} + b3 \times \text{mean uranium} + b4 \times \text{mean arsenic} + b5 \times \text{diabetes} + \text{offset } [\ln(\text{population})] \tag{2}$$

Mean nitrate and atrazine concentrations were categorized into two groups for each contaminant. Group 1 represented the watersheds with the lowest mean nitrate and atrazine concentrations, 6.94 mg/L and 0.00 µg/L, respectively. Group 2 represented all other watersheds included in the study.

Univariable analyses were conducted first between each independent variable and the outcome variable. The multivariable analysis model was built with the dependent variable and the predictors that showed a positive association in the univariable analysis.

## 3. Results

### 3.1. Descriptive Statistics

3.1.1. Birth Defects

From 1995 to 2014, 24,965 children born in Nebraska were diagnosed with at least one type of birth defect reported in the Nebraska Birth Defect Registry. A total count of 45,134 birth defect cases were reported. Birth defect prevalence per watershed ranged from 2.76 to 23.79 per 100 live births (Table 1).

**Table 1.** Birth defect prevalence per watershed in Nebraska during 1995–2014.

| Watershed Name | Prevalence (per 100 Live Births) |
|---|---|
| Beaver | 9.96 |
| Big Nemaha | 10.16 |
| Big Papillion-Mosquito | 10.91 |
| Blackbird-Soldier | 18.67 |
| Cedar | 6.18 |
| Frenchman | 6.84 |
| Harlan County Reservoir | 5.59 |
| Horse | 4.86 |
| Keg-Weeping Water | 9.39 |
| Keya Paha | 3.45 |
| Lewis and Clark Lake | 20.47 |
| Little Nemaha | 8.62 |

| | |
|---|---|
| Logan | 7.71 |
| Loup | 5.24 |
| Lower Elkhorn | 10.19 |
| Lower Little Blue | 5.44 |
| Lower Lodgepole | 2.76 |
| Lower Middle Loup | 7.76 |
| Lower Niobrara | 13.61 |
| Lower North Loup | 9.52 |
| Lower North Platte | 7.68 |
| Lower Platte | 18.57 |
| Lower Platte-Shell | 15.32 |
| Lower South Platte | 9.31 |
| Medicine | 9.07 |
| Middle Big Blue | 8.07 |
| Middle Niobrara | 5.51 |
| Middle North Platte-Scotts Bluff | 4.57 |
| Middle Platte-Buffalo | 6.55 |
| Middle Platte-Prairie | 7.88 |
| Middle Republican | 6.68 |
| Mud | 5.31 |
| Niobrara Headwaters | 3.45 |
| North Fork Elkhorn | 6.87 |
| Ponca | 23.79 |
| Red Willow | 16.02 |
| Salt | 8.74 |
| South Fork Big Nemaha | 9.74 |
| South Loup | 10.11 |
| Stinking Water | 8.56 |
| Tarkio-Wolf | 13.53 |
| Turkey | 10.13 |
| Upper Big Blue | 7.32 |
| Upper Elkhorn | 12.37 |
| Upper Little Blue | 4.62 |
| Upper Middle Loup | 3.33 |
| Upper Niobrara | 4.61 |
| Upper North Loup | 19.50 |
| Upper Republican | 5.46 |
| Upper White | 3.97 |
| West Fork Big Blue | 13.15 |
| Wood | 7.34 |

### 3.1.2. Groundwater Quality Data

Groundwater samples were collected from wells within the Lower Elkhorn, Upper Elkhorn, North Fork Elkhorn, Logan, Upper Big Blue, and West Fork Big Blue watersheds.

Anthropogenic agrichemical contaminants nitrate and atrazine were detected across the watersheds in varying concentrations. Groundwater nitrate concentrations ranged from 0.56 mg/L to 97.21 mg/L among the samples collected, with those highest in the Logan watershed region in Northeast Nebraska (Table 2). Groundwater atrazine concentrations ranged from 0.00 to 11.23 µg/L. Geogenic groundwater contaminants uranium and arsenic were detected in all of the watersheds. Groundwater uranium concentration ranged from 0.00 to 143.7 µg/L. Groundwater arsenic concentrations ranged from 0.38 to 19.84 µg/L (Table 2).

**Table 2.** Descriptive statistics for the groundwater quality data.

| Watershed | Samples (n) | Nitrate (mg/L) MCL = 10 | | | Atrazine (µg/L) MCL = 3 | | | Uranium (µg/L) MCL = 30 | | | Arsenic (µg/L) MCL = 10 | | |
|---|---|---|---|---|---|---|---|---|---|---|---|---|---|
| | | Mean | Min | Max | Mean | Min | Max | Mean | Min | Max | Mean | Min | Max |
| Logan | 20 | 21.37 | 1.18 | 97.21 | 0.00 | 0.00 | 0.00 | 8.02 | 0.11 | 28.03 | 2.61 | 0.40 | 8.04 |
| Lower Elkhorn | 37 | 9.09 | 1.14 | 24.58 | 0.01 | 0.00 | 0.29 | 9.81 | 0.06 | 83.08 | 3.17 | 0.40 | 10.51 |
| North Fork Elkhorn | 21 | 8.38 | 0.56 | 39.67 | 0.04 | 0.00 | 0.76 | 3.92 | 0.00 | 8.32 | 4.73 | 0.47 | 14.08 |
| Upper Elkhorn | 17 | 6.94 | 1.36 | 16.47 | 0.09 | 0.00 | 0.63 | 5.99 | 0.04 | 34.24 | 4.84 | 0.11 | 19.84 |
| Upper Big Blue | 17 | 18.24 | 0.73 | 58.35 | 0.18 | 0.00 | 1.39 | 13.18 | 0.55 | 143.7 | 3.61 | 1.22 | 6.13 |
| West Fork Big Blue | 20 | 16.12 | 0.87 | 55.97 | 1.11 | 0.00 | 11.23 | 6.33 | 0.24 | 15.03 | 3.14 | 0.33 | 5.04 |

Table 3 shows the percentages of samples above the MCLs for the water contaminants analyzed.

**Table 3.** Percentages of samples with analyte concentrations above their specific MCLs.

| | | Percent above MCL (%) | | | |
|---|---|---|---|---|---|
| Watershed | Samples (n) | Nitrate | Atrazine | Uranium | Arsenic |
| Logan | 20 | 50 (10/20) | 0.0 | 5 (1/20 | 0.0 |
| Lower Elkhorn | 37 | 41 (15/37) | 0.0 | 5 (2/37) | 3 (1/37) |
| North Fork Elkhorn | 21 | 19 (4/21) | 0.0 | 0.00 | 5 (1/21) |
| Upper Elkhorn | 17 | 24 (4/17) | 0.0 | 6 (1/17) | 12 (2/17) |
| Upper Big Blue | 17 | 65 (11/17) | 0.0 | 6 (1/17) | 0.0 |
| West Fork Big Blue | 20 | 65 (14/20) | 10 (2/20) | 0.0 | 0.0 |
| All | 132 | 43.9 (58/132) | 1.5 (2/132) | 3.8 (5/132) | 3.0 (4/132) |

Among the watersheds included in the study, 19% (North Fork Elkhorn) to 65% (West Fork Big Blue) of the domestic wells tested are in excess of the MCL of 10 mg/L for nitrate. While nitrate exceeds the MCL in approximately half of the domestic wells, groundwater atrazine concentration was only found above the MCL (3 mg/L) in 2 of 132 wells tested. Among domestic wells sampled in this study about 4% of the wells exceeded the drinking water MCL for uranium (30 µg/L), and 3% of the wells exceed the MCL of 10 µg/L for arsenic.

### 3.1.3. Correlations between Variables

All data points corresponding to nitrate, atrazine, uranium, and arsenic concentrations measured were used to conduct the analysis. Birth defect counts, tobacco use, and diabetes status were also included in the analysis (Table 4)

**Table 4.** Pearson correlation coefficients.

| | Nitrate | Atrazine | Uranium | Arsenic | Diabetes | Tobacco | Birth Defect Cases |
|---|---|---|---|---|---|---|---|
| Nitrate | 1 | 0.02 | 0.03 | −0.18 | −0.23 | −0.26 | −0.24 |
| Atrazine | 0.02 | 1 | −0.01 | −0.04 | −0.09 | −0.03 | 0.10 |
| Uranium | 0.03 | −0.01 | 1 | −0.08 | 0.03 | 0.01 | −0.02 |
| Arsenic | −0.18 | −0.04 | −0.08 | 1 | −0.06 | −0.01 | −0.01 |
| Diabetes | −0.23 | −0.09 | 0.03 | −0.06 | 1 | 0.96 | 0.76 |
| Tobacco | −0.03 | −0.03 | 0.01 | −0.01 | 0.96 | 1 | 0.88 |
| Birth defect cases | −0.24 | 0.10 | −0.02 | −0.01 | 0.76 | 0.88 | 1 |

The correlation analysis revealed high correlations between the covariates, diabetes, and tobacco ($r > 0.80$); thus, only one of these variables will be included in the statistical full model.

### 3.2. Geospatial Analysis

Figure 2 represents a map of Nebraska birth defect prevalence distribution per watershed in quartiles, with the first quartile prevalence (yellow color) below the national average. Watersheds not included in the study are left blank (white color on the map).

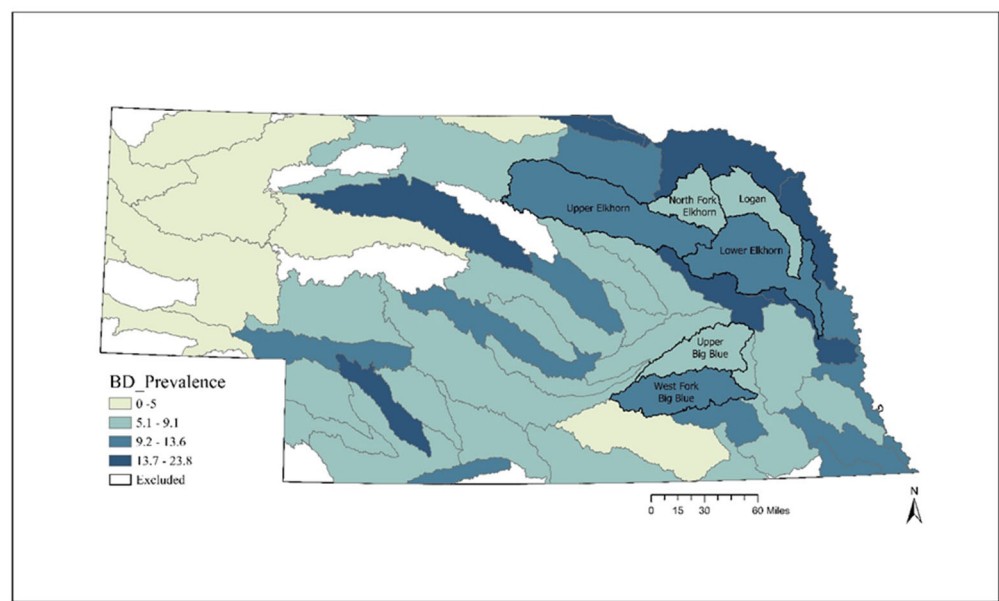

**Figure 2.** Birth defects prevalence in all Nebraska watersheds during 1995–2014.

More than 80% of Nebraska watersheds have birth defect prevalences that are above the national average (3 per 100 live births).

Figure 3 represents sampling locations with nitrate concentrations above 10 mg/L and birth defect (infant count) distributions per watershed region.

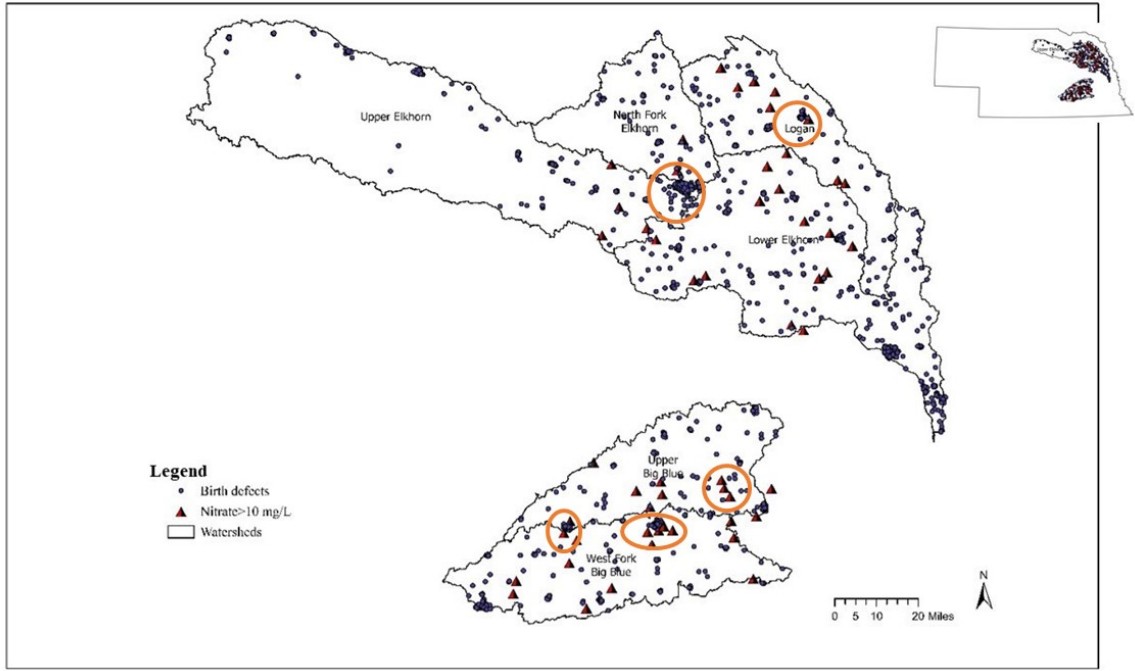

**Figure 3.** Geospatial representation of birth defect counts vs. nitrate concentration above 10 mg/L in the selected watersheds.

We observed that several clusters of birth defect locations also have high nitrate concentrations (above the MCL), as shown inside the circles on the map.

In the following map, birth defect counts were plotted against all sites with atrazine levels above 0.00 µg/L, including atrazine concentrations above the MCL (Figure 4).

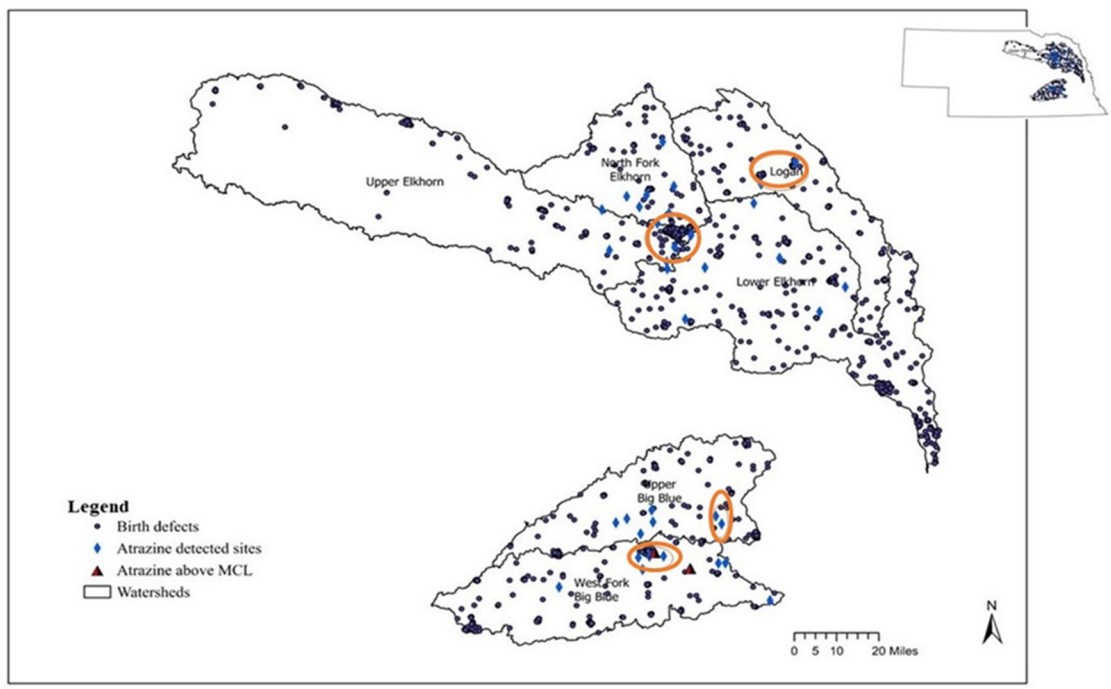

**Figure 4.** Geospatial representation of birth defect counts vs. atrazine concentration detected and above the MCL (3 µg/L) in the selected watersheds.

We observed that atrazine concentration was above 0.00 µg/L in many locations, with clusters of cases of birth defects, as indicated inside the circles on the map.

Figure 5 represents wells with arsenic concentrations above the MCL (10 µg/L) plotted against birth defect counts in the watersheds included in the study.

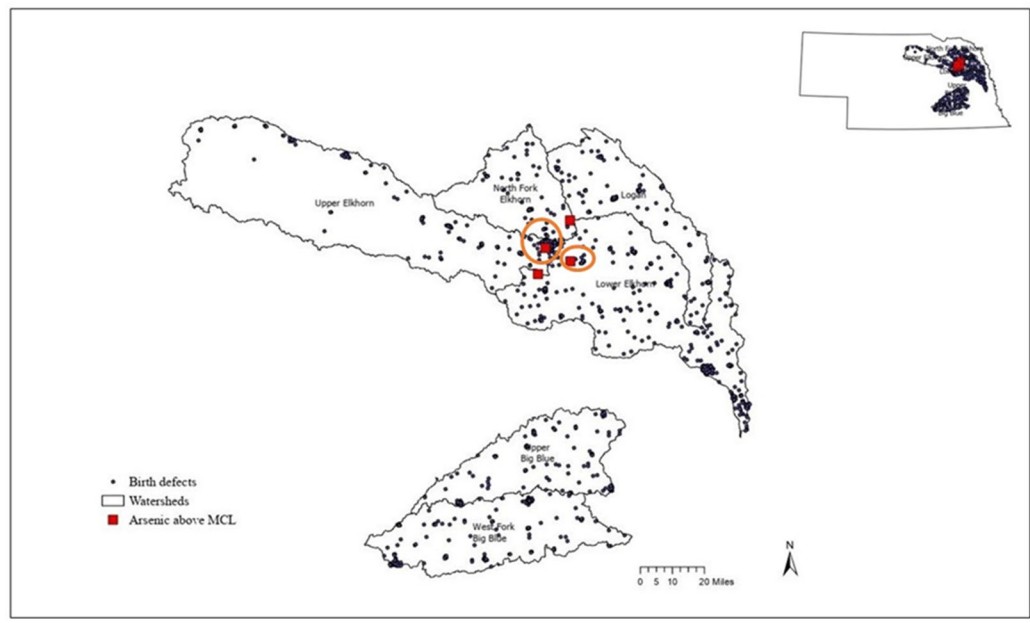

**Figure 5.** Geospatial representation of birth defect counts vs. arsenic concentration above 10 µg/L in the selected watersheds.

We observed clusters of birth defect locations with arsenic concentrations above the MCL, as shown inside the circles on the map.

*3.3. Inferential Statistics*

We conducted a negative binomial regression analysis in order to assess the association between birth defects and nitrate, atrazine, uranium, and arsenic concentrations.

Table 5 shows the results of the univariable and multivariable negative binomial regression analyses.

**Table 5.** Analysis of the associations between birth defects and nitrate and atrazine concentrations in drinking water.

| Variables | Birth Defects | |
| :---: | :---: | :---: |
| | Univariable Analysis IRR$_c$ (95% CI) [a] | Multivariable Analysis IRR$_a$ (95% CI) [a] |
| Nitrate | | |
| Group 1 | Reference | Reference |
| Group 2 | 1.44 (1.40–1.50) | 0.86 (0.83–0.89) |
| Atrazine | | |
| Group 1 | Reference | Reference |
| Group 2 | 2.84 (2.75–2.93) | 1.62 (1.56–1.70) |
| Diabetes | 1.03 (1.03–1.04) | 1.03 (1.02–1.04) |
| Tobacco [b] | 1.01 (1.01–1.02) | N/A |
| Uranium | 1.03 (1.02–1.03) | 0.96 (0.95–0.97) |
| Arsenic | 0.85 (0.84–0.86) | 0.92 (0.89–0.94) |

[a] Negative binomial regression analysis performed. The estimate of the model is IRR$_c$/IRR$_a$ = incidence rate ratio (crude/adjusted). In this study it is referred to as birth defects prevalence. [b] The variable tobacco was not included in the multivariable analysis, because it is collinear with the variable diabetes; only one of them should be included in the model.

3.3.1 Birth Defects and Nitrate

While the univariable analysis showed a positive association between nitrate concentrations above 6.94 mg/L and birth defects (IRR = 1.44; CI: 1.40–1.50), the multivariable analysis showed a negative association between the two (IRR = 0.86; CI: 0.83–0.89).

3.3.2 Birth Defects and Atrazine

In both the crude and adjusted models, watersheds with atrazine concentrations in group 2 have a higher prevalence of birth defects compared to the watersheds in group 1 (concentration above 0.00 µg/L).

3.3.3 Birth Defects and Uranium

The univariable analysis found a weak positive association between birth defects and uranium (IRR = 1.03; CI: 1.02–1.03), while the multivariable regression analysis showed a negative association (IRR = 0.96; CI: 0.95–0.97).

3.3.4 Birth Defects and Arsenic

In both the univariable (IRR = 0.85; CI: 0.84–0.86) and multivariable (IRR = 0.92; CI: 0.89–0.94) regression analyses, we observed a negative association between arsenic and birth defects.

**4. Discussion**

In this study, we determined the prevalence at birth of all birth defects in Nebraska watersheds. More than four-fifths of Nebraska watersheds reported a higher birth defect

prevalence (Table 1) than the national average of 5 cases per 100 live births [40]. Moreover, the watersheds included in the study had birth defect prevalence in the second and third quartiles (Figure 1), all above the national average.

These results aligned with the Nebraska Department of Health and Human Services report of a birth defect rate of 6 cases per 100 live births between 2011 and 2015 in Nebraska [3]. Additionally, Corley et al. [33] in their study revealed that Nebraska shared a disproportionate burden of birth defect-related deaths compared to the nation. Indeed in 2011, the mortality rate from birth defects was 1.94 per 100,000 in Nebraska compared to 1.27 per 100,000 for the United States.

We found relatively high nitrate mean concentrations in the watersheds that were included in the study. The mean nitrate concentrations ranged between 6.94 and 21.37 mg/L (Table 2). Three watersheds had groundwater nitrate concentrations above the MCL, and one is less than one unit lower than the MCL. Those concentrations correspond, as mentioned above, to watersheds with birth defect prevalences that are above the national average.

We also observed in the geospatial analysis that areas with clusters of birth defects were more likely to have high contaminant levels of nitrate (Figure 3), atrazine (Figure 4), and arsenic (Figure 5), suggesting a spatial relationship between these contaminants and the birth defect prevalence.

In our univariable analysis, we found that watersheds with nitrate concentration above 6.94 mg/L had higher prevalence of birth defects compared to the reference group (Table 5. In the multivariable analysis, after controlling for the covariates of arsenic and uranium concentrations and a maternal risk factor (diabetes), we found a negative association between nitrate concentration and birth defect incidence rate. This observation can be explained by the fact that when additional birth defect risk factors are combined, higher nitrate concentration resulted in miscarriages, giving the impression that the number of birth defects has decreased; instead, the higher nitrate concentration was lethal in utero. This finding corroborates a series of four case studies that suggested an association between maternal exposure to high nitrate levels in drinking water and the occurrence of spontaneous abortions [41].

The findings of the current research corroborate many other studies that reported a positive association between relatively higher nitrate concentration in drinking water and the occurrence of birth defects. Among them, Brender and Weyer [12] described a study that observed that women with a drinking water nitrate concentration above 10 mg/L had four times the odds (95% CI: 1.0–15.4) of a birth defect (anencephaly) than their counterparts whose water supplies had a nitrate concentration at or below 10 mg/L.

Additionally, another study found a significant increase (OR = 2.44; CI:1.05–5.66) in the incidence of birth defects for drinking water with nitrate levels of 1–5.56 mg/L compared to <1 mg/L, after controlling for variables such as the infant's gender, season of birth, the mother's age and parity, and some maternal risk factors (smoking, diabetes, and thyroid disease) [11].

Moreover, maternal ingestion of drinking water with a nitrate concentration at or above 5 mg/L in the periconceptual period was associated with greater odds of birth defects (spina bifida) compared to mothers whose drinking water had a nitrate level below 0.91 mg/L, after adjusting for confounders such as the mother's age, ethnicity, education level, and folic acid intake (OR = 2.0; 95% CI: 1.3–3.2) [13].

We also found that in both the univariable and adjusted models that atrazine levels > 0.00 μg/L were associated with higher incidence rates of birth defects. Atrazine is potentially teratogenic because of its abilities to cause oxidative stress and congenital anomalies that were observed in animal models [42,43]. The correlation between atrazine in drinking water and birth defects occurrence has been reported in many other studies. Indeed, Mattix et al. [44] in their study that used water quality data from USGS and birth defect data from the CDC and the Indiana Department of Health observed that increased monthly atrazine concentrations in surface water are correlated with higher rates of

abdominal wall defects [44]. Similarly, a positive association was found between gastroschisis and maternal proximity to water quality monitoring sites reporting atrazine concentrations > 3 μg/L [10].

Moreover, Agopian et al. [9] observed that maternal exposure to medium-low to medium levels of atrazine during the prenatal period was associated with birth defects—male genital malformations—in Texas during 1999–2008.

The univariable analysis revealed a weak positive association between birth defects and uranium concentrations in drinking water (Table 5). This result aligns with many other studies that suggested a positive relationship between uranium in drinking water and the occurrence of birth defects [27–29]. In animal models, a study reported that very high concentrations of uranium in the drinking water resulted in birth defects and a rise in fetal deaths [45].

We observed a negative association between uranium and birth defects in the multivariable analysis, as well as between arsenic and birth defects both in the univariable and multivariable analyses (Table 5). While several studies have shown a positive association between birth defects and arsenic [12,30–32], a negative association has been reported as a spontaneous pregnancy loss resulting from exposure to arsenic in drinking water by many studies [46].

The explanation for these findings may be that when multiple risk factors of birth defects are combined, they result in pregnancy loss, which may appear in the analysis as if the contaminants were protective of birth defects.

It is necessary to recognize that some studies that investigated the relationship between birth defects and nitrate or atrazine did not find any associations. Such studies include a case-control study conducted in Washington state that observed no association between gastroschisis and mothers living close to a water quality monitoring site that reported nitrate concentrations above the MCL (10 mg/L) [10]. Additionally, Mattix et al. [44], who studied the relationship between nitrate and birth defects in Indiana, found no correlation between monthly abdominal wall defect rates and higher nitrate concentrations in drinking water. Moreover, researchers in Texas conducted a case-control study that observed no positive association between maternal exposure to high atrazine levels in drinking water and an increased risk of congenital heart defects [47].

Compared to our study, the above-mentioned studies—that did not find any associations between birth defects and nitrate or atrazine concentrations—were regarding specific birth defect types (abdominal wall defect or congenital heart defects). It is possible that if other birth defect types were investigated, a correlation could have been found. Our research looked at the relationship between all birth defects and nitrate or atrazine levels in drinking water.

*Study Strengths and Limitations*

This study had the advantage of using a large data set of birth defects that occurred in Nebraska during 1995–2014. Many risk factors for birth defects were controlled for in this study: arsenic, uranium, and maternal diabetes status. However, as an ecological study, individual-level exposure was not measured. Additionally, it was assumed that current contaminant levels were similar to levels during the exposure period. This assumption is supported by the agricultural activities and practices that have been steady or increasing for many decades in Nebraska. Moreover, previous findings showed that nitrate and atrazine concentrations measured in monitoring wells have not dramatically changed in 1987–2016 (the mean estimate changed by less than 5%) [48].

## 5. Conclusions

In this study, we determined the relationship between birth defects and nitrate, atrazine, uranium, and arsenic concentrations in drinking water in selected Nebraska watersheds.

The prevalence at birth of birth defects was calculated for each Nebraska watershed during 1995–2014. The mean nitrate, atrazine, uranium, and arsenic concentrations were computed for the watersheds included in the study. The findings showed that birth defects prevalence was above the national average for the watersheds included in the study. Mean nitrate concentrations were relatively high (6.94–21.37 mg/L), and atrazine was detected at concentrations above 0.00 µg/L in all but one of the watersheds included in the study. In the regression analysis, a positive association was found between higher levels of nitrate in drinking water and the prevalence of birth defects. Similarly, compared to watersheds with atrazine levels at 0.00 µg/L, watersheds with higher levels of atrazine showed elevated prevalences of birth defects. Uranium and arsenic were negatively associated with birth defects. This study suggests that chronic exposure to nitrate and atrazine concentrations below the maximum contaminant limits may result in birth defects, whereas exposure to uranium and arsenic increase the risk of pregnancy loss. It also highlights the relationship between anthropogenic activities, agriculture practices, as well as geogenic release of elements as well as water contamination, with adverse health effects in children.

Recommendations to address the health effects of water contamination include the following: (1) continuous monitoring of private well water by state jurisdictions to ensure contaminants are present at safe levels; (2) provision of adequate water filtration systems or alternate sources of water for households with high contaminant levels in drinking water; and (3) implementation of efficient agricultural practices to improve soil uptake and reduce agrichemical runoff into water supplies.

**Author Contributions:** Conceptualization, B.S.O. and E.G.R.; methodology, B.S.O. and M.Z.; software, B.S.O.; validation, B.S.O. and E.G.R.; formal analysis, B.S.O.; resources, E.G.R. and K.A.W.; data curation, B.S.O. and F.I.R.; writing—original draft preparation, B.S.O.; writing—review and editing, B.S.O., E.G.R., M.Z., S.L.B.-H., K.A.W. and F.I.R.; visualization, B.S.O.; supervision, E.G.R.; funding acquisition, B.S.O. and E.G.R. All authors have read and agreed to the published version of the manuscript.

**Funding:** This research was funded by the Centers of Disease Control and Prevention/National Institute for Occupational Safety and Health through the Co-operative Agreement U54 OH010162 to the Central States Center for Agricultural Safety and Health (CS-CASH); [U54 OH010162]. Support was also provided by the Robert B. Daugherty Water for Food Global Institute at the University of Nebraska, and the Child Health Research Institute in Omaha, Nebraska.

**Institutional Review Board Statement:** The study was conducted according to the guidelines of the Declaration of Helsinki, and approved by the Institutional Review Board of the University of Nebraska Medical Center, NE, USA (protocol code 414-16-EP and date of approval: 7 July 2016).

**Informed Consent Statement:** Informed consent was obtained from participants to provide drinking water samples for testing.

**Data Availability Statement:** Birth defect and live birth data are not available to the public or research community unless access is granted by the Nebraska DHHS upon completion and approval of data use agreement forms. For further information contact the Nebraska DHHS at DHHS.PublicRecords@nebraska.gov.

**Acknowledgments:** We extend our gratitude to the Lower Elkhorn and Upper Big Blue natural resource districts for their support in water sample collections. We also thank Taylor Rosso in Weber's laboratory for analyzing some samples for nitrate concentrations.

**Conflicts of Interest:** The authors declare no conflicts of interest.

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
