# Peer review of "Investigation of a Possible Relationship between Anthropogenic and Geogenic Water Contaminants and Birth Defects Occurrence in Rural Nebraska"

_water, doi:10.3390/w14152289_

Round 1
Reviewer 1 Report
Dear Authors,
I am glad having an opportunity to review the manuscript. It is a interesting paper. I particularly appreciate the main idea of this paper, information that chronic exposure to the selected waterborne contaminants below the maximum contaminant levels may result in birth defects. It also highlighted the relationship between anthropogenic activities (agriculture practices), water contamination, and adverse health effects on children. Authors recommended to support these findings so that regulations can be implemented in the form of continuous monitoring of water in private wells and improvement of agricultural practices, it is very important in this moment.
While reading the submitted manuscript i see only several questions on editorial issues.
1. Are table were made in accordance with the guidelines for authors? Attention it concerns the size of the descriptions.
2. Discussion: Did the authors find similar studies in the literature, with which to compare the results of their study?
Author Response
Dear reviewer,
We appreciate your feedback on our manuscript. We strived to address your comments in this version of the paper.
Response to Reviewer 1 Comments
Point 1: Are table were made in accordance with the guidelines for authors? Attention it concerns the size of the descriptions.
Response 1: Yes, we reviewed the instructions for authors section, and the tables are in accordance with the guidelines provided.
Point 2. Discussion: Did the authors find similar studies in the literature, with which to compare the results of their study?
Response 2: Yes, in the discussion section, we mentioned studies in the literature that found similar results. This can be found in lines 330 to 370.
Reviewer 2 Report
Journal: Water
Title: Investigation of a possible relationship between anthropogenic and geogenic water contaminants and birth defects occurrence in rural Nebraska
The authors investigate the connection of the use of drinking water from the Nebraska area, which contains, as the authors state, 4 contaminants (nitrate, atrazine, uranium and arsenic) with birth defects. They noticed that nitrate is one of the main links with birth defects. The research is based on birth defects data from 1995 to 2014 and analyses of the water used by the population from 2021 and 2022. My main question is the connection between the results of the water analysis that was carried out 6 years after the last birth defects data. You should have the results of water analysis from the same period 1995-2014. The authors concluded that even today the concentrations of the 4 mentioned contaminants are high, but the question is whether they have increased or decreased compared to the period 1995-2014. Therefore, I find the whole research questionable considering the comparability of data from different time periods.
I wonder what the situation is today, in 2022, whether the number of birth defects has decreased since the problem is more than worrying.
Reading the paper and looking from the point of view of ecotoxicology, of the mentioned 4 contaminants, nitrate is the least toxic, however in lines 317-318 you explain the interaction of nitrate and the formation of a new compound. I think this should be mentioned in the introduction section.
SPECIFIC COMMENTS
Line 26: „…contaminants below the maximum…“ – suggestion: contaminants even below the maximum.
In the introductory part, you should refer to:
- geogenic source of arsenic and uranium
- why the mentioned 4 contaminants were taken into account
- are there any studies on the complete water analysis, that is, why these 4 contaminants were chosen
- have you conducted a complete water analysis, because the question arises for the reader, why exactly these 4 contaminants...
Lines 126-131: The writing style, "nitrate samples, uranium samples" it seem like you took a sample that only contains uranium or nitrate. Better to express: Sample for uranium (nitrate) analysis, ...
Line 144: Do not use abbreviations in the title, write the full name
Line 216: „Nineteen (19) to 65% of the“ – I don't understand the meaning.
Line 222: 3.13. change to 3.1.3.
Line 397, 400: „0 ug/L“- write as before, to two decimal places
Pay attention to the legibility of the letters inside the Figures.
Author Response
Dear reviewer,
We appreciate your feedback on our manuscript. We strived to address your comments in this version of the paper.
Response to Reviewer 2 Comments
Point 1: The authors investigate the connection of the use of drinking water from the Nebraska area, which contains, as the authors state, 4 contaminants (nitrate, atrazine, uranium and arsenic) with birth defects. They noticed that nitrate is one of the main links with birth defects. The research is based on birth defects data from 1995 to 2014 and analyses of the water used by the population from 2021 and 2022. My main question is the connection between the results of the water analysis that was carried out 6 years after the last birth defects data. You should have the results of water analysis from the same period 1995-2014. The authors concluded that even today the concentrations of the 4 mentioned contaminants are high, but the question is whether they have increased or decreased compared to the period 1995-2014. Therefore, I find the whole research questionable considering the comparability of data from different time periods.
Response 1: Thank you for raising an interesting point. Yes, we agree that the use of exposure data (water quality data) for the same period as the health outcome data was ideal. However, our choice of using recently collected water quality data is based on several reasons:
- There are no publicly available water quality data from domestic drinking well water for the period from 1995 - 2014. This is explained by the fact that the safe drinking water act enacted by the congress in 1974 does not regulate private well water. So, the private wells are not regularly tested for contaminants levels, it is up to the well owner to assure the quality of their drinking water.
- In a previous study [48] our findings showed that nitrate and atrazine concentrations measured in monitoring wells have not dramatically changed from 1987 - 2016 (the mean estimate changed by less than 5%). This can be explained by the steady agricultural practices in Nebraska for decades, practices that are the main sources of atrazine and nitrate in water in farming areas as described in the literature. We updated the text and added a reference in lines 397 – 399.
Point 2: I wonder what the situation is today, in 2022, whether the number of birth defects has decreased since the problem is more than worrying.
Response 2: We would like to determine the prevalence of birth defect in more recent years in Nebraska, however, we requested and was denied access to recent health data from the Nebraska department of health and human services because of a recent Nebraska legislation.
Point 3: Reading the paper and looking from the point of view of ecotoxicology, of the mentioned 4 contaminants, nitrate is the least toxic, however in lines 317-318 you explain the interaction of nitrate and the formation of a new compound. I think this should be mentioned in the introduction section.
Response 3: We moved those lines to the introduction as suggested.
SPECIFIC COMMENTS
Point 4: Line 26: „…contaminants below the maximum…“ – suggestion: contaminants even below the maximum.
Response 4: we considered your suggestion.
Point 5: In the introductory part, you should refer to:
- geogenic source of arsenic and uranium
- why the mentioned 4 contaminants were taken into account
- are there any studies on the complete water analysis, that is, why these 4 contaminants were chosen
- have you conducted a complete water analysis, because the question arises for the reader, why exactly these 4 contaminants...
Response 5:
- geogenic source of arsenic and uranium was added in the introduction.
- The 4 contaminants were chosen because:
. Nebraska is an agricultural state with the extensive use of nitrate-containing fertilizers and pesticides such as atrazine. These agrichemicals have been found in previous studies at high concentrations in certain locations in Nebraska.
. Higher nitrate concentrations lead to the mobilization of geogenic uranium resulting in water contamination. Higher uranium and arsenic concentrations have been found in Nebraska.
To summarize, the four contaminants were chosen because water contamination at higher concentrations with nitrate, atrazine, uranium, and arsenic is frequent in Nebraska which is mostly related to the agricultural practices in the state.
Lines 64-86 explained why the 4 contaminants were chosen. References are added.
Point 6: Lines 126-131: The writing style, "nitrate samples, uranium samples" it seem like you took a sample that only contains uranium or nitrate. Better to express: Sample for uranium (nitrate) analysis, ...
Response 6: the wording has been fixed
Point 7: Line 144: Do not use abbreviations in the title, write the full name
Response 7: the title has been updated with the full name
Point 8: Line 216: „Nineteen (19) to 65% of the“ – I don't understand the meaning.
Response 8: As shown in Table 3, 19% of samples tested for nitrate in the North Fork Elkhorn watershed were above the MCL, and 65% of samples tested for nitrate in West Fork Big Blue were above the MCL. We wanted to show the range of samples above the MCL among the watersheds included in the study.
The wording has been updated in the text for more clarity.
Point 9: Line 222: 3.13. change to 3.1.3.
Response 9: It has been corrected.
Point 10: Line 397, 400: „0 ug/L“- write as before, to two decimal places
Response 10: It has been fixed.
Round 2
Reviewer 2 Report
Since the Authors provided all the answers to the questions and made changes to the manuscript, I accept the manuscript in this form.